# Spectrofluorimetric Analysis of Riboflavin Content during Kombucha Fermentation

**DOI:** 10.3390/biotech13020020

**Published:** 2024-06-11

**Authors:** Mojca Čakić Semenčić, Adrianna Biedrzycka, Anna Kiczor, Sunčica Beluhan, Filip Šupljika

**Affiliations:** 1Department of Chemistry and Biochemistry, Faculty of Food Technology and Biotechnology, University of Zagreb, 10000 Zagreb, Croatia; mcakic@pbf.unizg.hr (M.Č.S.); sunbel@pbf.hr (S.B.); 2Faculty of Chemistry, Maria Curie Skłodowska University, 20-031 Lublin, Poland; a.biedrzycka@poczta.umcs.lublin.pl (A.B.); a.kiczor@poczta.umcs.lublin.pl (A.K.)

**Keywords:** kombucha, riboflavin, vitamin B2, spectrofluorimetry

## Abstract

Kombucha is a traditional beverage obtained by the microbial fermentation of tea using a symbiotic culture of bacteria and yeasts. In addition to several documented functional properties, such as anti-inflammatory activity and antioxidant activity, kombucha is often credited with high levels of vitamins, including riboflavin. To our knowledge, the vitamin B2 content in traditionally prepared kombucha has been determined in only two studies, in which the concentration measured by the HPLC technique ranged from 2.2 × 10^−7^ to 2.1 × 10^−4^ mol dm^−3^. These unexplained differences of three orders of magnitude in the vitamin B2 content prompted us to determine its concentration during the cultivation of kombucha under very similar conditions by spectrofluorimetry. The B2 concentrations during the 10-day fermentation of black tea ranged from 7.6 × 10^−8^ to 3.3 × 10^−7^ mol dm^−3^.

## 1. Introduction

Kombucha is a traditional Chinese beverage created by cultivating a symbiotic culture of bacteria and yeast on sweetened tea. Due to its proven health benefits [1,2,3,4,5,6] as well as its ability to ferment on different substrates [7,8,9], resulting in different flavors, kombucha is now popular worldwide. The most abundant microorganisms in kombucha are the genera *Acetobacter* and *Gluconobacter* and yeasts, mainly from the genera *Brettanomyces* and *Zygosaccahromyces* [10]. Although green and black tea are the traditional substrates for kombucha, it is now also grown on many other alternative raw materials [8], such as soy [11], juice [7], herbal extracts [12], milk [9], etc. Sucrose at a concentration of 1–20% is the most commonly used carbon source for kombucha fermentation, although other sugars, such as lactose, glucose, fructose [13], or mollases [14], can also be used. The chemical composition of this beverage depends on the microbial population introduced into the fermentation medium, the substrate, the type and concentration of sugar added, and the growth parameters: temperature, pH, oxygen, and fermentation time [15]. The chemical compounds commonly found in kombucha include various organic acids, such as acetic acid when sucrose is used as the primary carbon source, sugars, ethanol, polyphenols, minerals, vitamins, anions, biogenic amines, purines, pigments, lipids, etc. [2,10,16]. Some of these chemical constituents are metabolic products formed during the fermentation process, others are derived from the substrate itself, and still others are formed by changing the structure of the originally introduced compounds during fermentation. The most abundant vitamins in kombucha produced with conventional substrates are the water-soluble vitamin C and the B-complex vitamins (B1, B2, B6, and B12). Since vitamins C, B1, B6, and B12 are present in relatively high concentrations, their quantification is straightforward and usually performed by routine UV spectrophotometric methods [17,18]. As for the content of vitamin B2, to our knowledge, only two papers [18,19] have been published on the quantification of its concentration in traditionally prepared kombucha. Using the HLPC apparatus, Malbaša et al. [19] detected 2.2 × 10^−7^ mol dm^−3^ of vitamin B2 in kombucha prepared with native inoculum and black tea sweetened with 7% sucrose after 10 days of fermentation. A slightly higher B2 concentration was found in kombucha grown on green tea under the same conditions. In the second more comprehensive study [18], the same author performed a similar experiment and reported B2 concentrations that were three orders of magnitude higher than those reported above. They were determined by HPLC and ranged from 1.1 × 10^−4^ to 2.1 × 10^−4^ mol dm^−3^ on the tenth day of fermentation.

Given the lack of literature data on the vitamin B2 concentration in kombucha, as well as the inexplicably large differences in the vitamin B2 content of kombucha grown under very similar conditions, the aim of this study was to determine the vitamin B2 concentration in kombucha grown on black tea using spectrofluorimetry. We chose this method because it is exceptionally sensitive, selective, inexpensive, rapid, and enables the detection of fluorescent analytes, such as water-soluble vitamins, in complex matrices [20,21]. Since kombucha is often touted as rich in vitamins, including B2, it is important to resolve the dilemma regarding its content in the traditionally prepared beverage.

## 2. Materials and Methods

### 2.1. Kombucha Preparation and Fermentation

Samples consisted of a local kombucha culture, grown in the laboratory for biochemical engineering, industrial microbiology, and technology of beer and malt (Faculty of Food Technology and Biotechnology, Zagreb, Croatia), cultivated on black tea (Franck d.d., Zagreb, Croatia) sweetened with sucrose (Viro d.d., Virovitica, Croatia). The substrate was prepared by infusing black tea with a total concentration of 9 g dm^−3^ in 400 mL of boiled tap water containing 7%, 9%, or 10% sucrose. After removing the tea bags, the solutions were cooled to room temperature and 3% (*m*/*V*) of the kombucha culture was added to the infusions along with 10% (*V*/*V*) of the mother liquor. The blank sample was prepared in the same manner, except for the addition of the last two ingredients. The wide glass jars were covered with sterile gauze (Lola Ribar d.d., Zagreb, Croatia) and left at room temperature for 10 days for fermentation. Samples were taken on days 1, 3, 6, 7, 8, 9, and 10 and filtered through 0.45 µm membranes (Macherey-Nagel, Düren, Germany) before B2 analysis.

### 2.2. Determination of B2 Content

The concentration of vitamin B2 was determined using PerkinElmer LS 55 fluorimeter equipped with 10 mm quartz cells. All measurements were performed at room temperature (25 ± 0.2 °C) using an excitation wavelength of 444 nm and an emission wavelength of 523 nm, ex. slit of 15 nm and em. slit of 15 nm. The scan speed was 200 nm min^−1^. Acetate buffer pH = 6, used for dilution of standard and samples, was prepared according to the procedure described by Pesez and Bartos [22]. All chemicals used for the preparation of the buffer were purchased from Merck (Darmstadt, Germany) and used without further purification. The pH values were measured using a digital Metrohm 913 pH meter. The calibration curve was constructed in the concentration range of 2–5 × 10^−7^ mol dm^−3^ by successive dilution of the standard solution of riboflavin (Sigma-Aldrich, Burlington, NJ, USA). The linear regression obtained was
*y* = 5.84 × 10^8^
*x* − 1.08(1)
with the correlation coefficient (*R*^2^) 0.99. Before the determination of riboflavin concentration, the filtered kombucha samples were diluted ten times with acetate buffer and their pH was determined. Three samples were taken on each measurement day, and all measurements were performed in triplicate.

## 3. Results and Discussion

A local kombucha culture was grown for 10 days on black tea containing 7%, 9%, or 10% sucrose. The quantitative determination of the riboflavin in the periodically collected samples was performed by a spectrofluorimetric method. The calibration curve of the riboflavin is shown in Figure 1 and proved to be linear, with a correlation coefficient of 0.99 in a concentration range of 2 × 10^−7^–5 × 10^−7^ mol dm^−3^. 

Before the determination of the riboflavin concentration, the filtered kombucha samples were diluted ten times with acetate buffer and their pH was determined. The pH of the three kombucha samples (Table 1) decreased from an initial value of 3.72 ± 0.01 to values in the range of 2.78 to 2.89 during the 10-day incubation, which was due to the formation of organic acids [23]. Since kombucha is an acidic beverage that generally has a pH between 2.5 and 3.5, and the fluorescence quantum yield of riboflavin is highest in the range between pH = 4 and pH = 9 and is pH-independent [24], an acetate buffer of pH = 6 was used for each measurement of the B2 concentration. 

After the first day of fermentation, the B2 concentrations ranged from 2.5 to 3.3 × 10^−7^ mol dm^−3^, whereas the blank sample contained 2.1 × 10^−7^ mol dm^−3^ B2. During cultivation, the fluorometrically determined riboflavin concentration in all three samples varied between about 1.0 × 10^−7^ and 3.0 × 10^−7^ mol dm^−3^ (Figure 2). These variations in vitamin B2 content during kombucha fermentation have already been observed during the cultivation on soymilk [25], but, because it is a different substrate, the absolute concentration of the vitamin B2 cannot be compared with our results. The variations observed in the vitamin B2 concentration may be due to the changes in the pH value during the kombucha fermentation. Indeed, at a pH above 4, the fluorescent neutral form of riboflavin dominates, but, when the pH drops below 4, the chemical equilibrium shifts towards the cationic form, which is non-fluorescent [24]. Since we carried out the measurements immediately after sampling, it could be that, despite using a buffer with a pH of 6, a new equilibrium, in which the fluorescent neutral form of riboflavin predominates, was not fully established. Since the blank itself, i.e., the tea, contains 2.1 × 10^−7^ mol dm^−3^ of riboflavin, we do not believe that the fluctuations in the content were caused by the leakage of inactive cells and the release of riboflavin into the kombucha. These two assumptions also explain the fact that the riboflavin concentration in all the samples on the last day of fermentation was lower than in the blank sample. The vitamin B2 content measured on the last day of fermentation was 1.4 × 10^−7^, 1.3 × 10^−7^, and 7.6 × 10^−8^ mol dm^−3^ for the samples containing 7%, 9%, and 10% sucrose, respectively, while the blank sample contained 1.7 × 10^−7^ mol dm^−3^ of B2. In view of the chemical structure of riboflavin, we assume that it does not interact with other kombucha components at these concentrations, which could lead to fluctuations in the content. It should also be noted that the fluorimetrically determined content of vitamin B2 is in agreement with that determined by Malbaša et al. [19], while it differs by three orders of magnitude from that determined by the same author a few years later [18], although, in all the cases, the cultivation took place under approximately the same conditions.

Since there are only two datasets in the literature on the content of vitamin B2 in traditional kombucha, which differ significantly from each other, the aim of this work was to quantify the riboflavin content during the 10-day fermentation of black tea. Using the spectrofluorimetric method, the content of vitamin B2 was determined to be between 7.63 × 10^−8^ and 3.3 × 10^−7^ mol dm^−3^, from which it can be concluded that B2 is present in kombucha in much lower concentrations compared to other water-soluble vitamins. Although it is often claimed that kombucha is rich in B2, compared to other common beverages, its riboflavin content is in the range of the concentration found in beer, wine, and other alcoholic beverages, where the concentration is between ~1 × 10^−6^ and ~1 × 10^−7^ mol dm^−3^, which is lower than the content in energy drinks, milk, orange juice, etc. (~1 × 10^−5^ mol dm^−3^ or even more) [26,27]. Given the lack of literature data on the vitamin B2 concentration in kombucha, as well as the inexplicably large differences in the B2 content of kombucha grown under very similar conditions, the objective of this study was to resolve the dilemma regarding the B2 concentration in the traditionally prepared beverage.

## Figures and Tables

**Figure 1 biotech-13-00020-f001:**
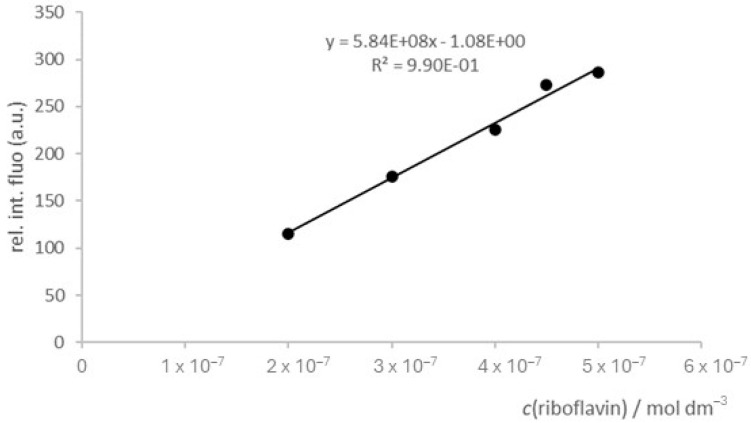
Calibration curve of riboflavin.

**Figure 2 biotech-13-00020-f002:**
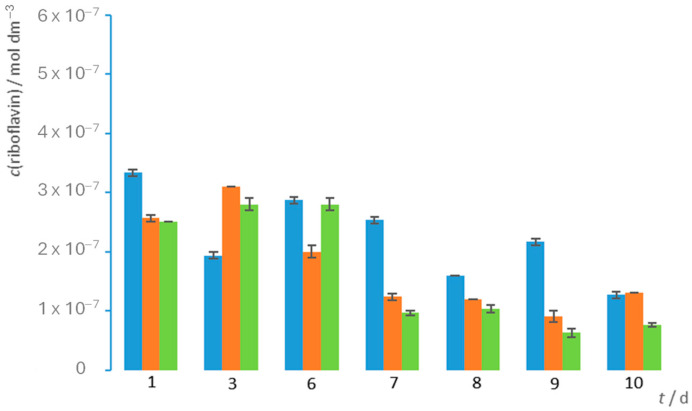
Changes in riboflavin content during fermentation of black tea. Riboflavin concentration during 10-day kombucha cultivation at sucrose mass concentration of 70 g dm^−3^ is shown in blue, at sucrose mass concentration of 90 g dm^−3^ in orange, and at sucrose mass concentration of 100 g dm^−3^ in green.

**Table 1 biotech-13-00020-t001:** pH during kombucha fermentation.

*γ*(Sucrose)/g dm^−3^			
Day	70	90	100
0	3.73	3.72	3.71
1	3.8	3.76	3.75
3	3.48	3.32	3.35
6	3.29	3.15	3.18
7	3.08	2.88	3.03
8	3	2.84	2.94
9	2.98	2.82	2.93
10	2.89	2.78	2.89

## Data Availability

The data that support the findings of this study are available on request from the corresponding author.

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
