# Peer review of "Spectrofluorimetric Analysis of Riboflavin Content during Kombucha Fermentation"

_biotech, 2024, doi:10.3390/biotech13020020_

Round 1
Reviewer 1 Report
Comments and Suggestions for Authors
Greetings, Authors
After an assessment, your work titled "Spectrofluorimetric Analysis of Riboflavin Content During Kombucha Fermentation" needs major adjustments before it can be considered for publishing in the BioTech Journal, even though it provides an intriguing topic.
The manuscript's present lack of emphasis on the spectrofluorimetric analysis itself is my main worry. The riboflavin content of kombucha is an interesting study, but the publication should go into more detail regarding the methodology and validation of the selected analytical approach.
Consequently, I suggest making the following changes:
1. Describe the Spectrofluorimetric Methodology in detail. Give a thorough explanation of the spectrofluorimetric technique you employed, along with the precise wavelengths at which riboflavin was excited and emitted throughout your experiment. This will make it easier for readers to comprehend and carry out your study.
2. Detailed Method Optimization: Provide a separate section outlining the spectrofluorimetric method's optimization procedure. Explain the modified parameters (such as solvent, pH effect, temperature, slit width, and wavelength) and provide an explanation for the final circumstances that were selected.
3. Include Figures for Spectral Data: It is essential to represent things visually. Provide figures that illustrate the riboflavin's actual excitation and emission spectra that you were able to collect throughout your study. This will not only validate your findings but also show how successful the approach is.
4. Include Statistical Analysis: Provide a solid statistical analysis of the data to support your conclusions. Report data on the riboflavin content during the fermentation process, including the mean, standard deviation, and variation. To determine the importance of changes over time, think about using statistical tests.
5. Validate the Spectrofluorimetric Method: Carefully record the method's validation. Information on specificity, accuracy, repeatability, and limit of detection (LOD) should be included. Describe any reference standards and calibration curves that were used for validation.
6. Contrast with Earlier Research: Give a thorough comparison of your results with earlier research that measured riboflavin in kombucha using HPLC. Talk about the possible benefits and drawbacks of spectrofluorimetry in contrast to HPLC.
7. The riboflavin's fluorescence quantum yield: It would be advantageous to look into the riboflavin's fluorescence quantum yield in your particular experimental setup. The dependability of your results will be strengthened and your spectrofluorimetric method will be further validated by comparing your measured quantum yield with the established theoretical value.
By implementing these changes, your work will be more aligned with BioTech Journal's emphasis on biotechnological approaches and applications. It will also improve the scientific rigor and effect of your research into kombucha fermentation and nutrient analysis.
Author Response
1. Describe the Spectrofluorimetric Methodology in detail. Give a thorough explanation of the spectrofluorimetric technique you employed, along with the precise wavelengths at which riboflavin was excited and emitted throughout your experiment. This will make it easier for readers to comprehend and carry out your study.
2. Detailed Method Optimization: Provide a separate section outlining the spectrofluorimetric method's optimization procedure. Explain the modified parameters (such as solvent, pH effect, temperature, slit width, and wavelength) and provide an explanation for the final circumstances that were selected.
We thank you for your careful reading of the manuscript, your comments and suggestions. First of all, we would like to emphasize that this paper is written in the form of a Communication in which the main experimental details are given to allow a repetition of the experiment (subsection 2.2. Determination of B2 content). The large differences in riboflavin concentration (three orders of magnitude) in almost identically grown kombucha published by the same author (Malbaša et al. 2004 and Malbaša et al. 2011) inspired us to conduct this study. When we got the results of our study, which were in good agreement with those of Malbaša et al. 2004, we contacted the author, he replied that he assumes that the high riboflavin concentrations in Malbaša et al. 2011 were the result of lysis of the cell culture due to the long storage of the samples in the freezer. Unfortunately, the results of this study are cited in numerous articles and reports published later.
3. Include Figures for Spectral Data: It is essential to represent things visually. Provide figures that illustrate the riboflavin's actual excitation and emission spectra that you were able to collect throughout your study. This will not only validate your findings but also show how successful the approach is
When planning our experiment, we used the information available in the literature about the excitation wavelength of riboflavin, 444 nm (Ahmad et al., Luminescence 2018, 33, 1070), which is close to the wavelength at which Malbaša performed the excitation (450 nm). Our emission maximum was at 523 nm, also close to the wavelength reported by Malbaša (530 nm). In the continuation of the measurement, we did not record the entire spectra, but only read the emission at 523 nm after excitation at 444 nm. We attach the emission spectrum (see attachment) and point out in particular that it was recorded at ex. slit and em. slit 15 nm (as indicated in the methods), which we had to use due to the low riboflavin concentrations in the samples and which is therefore of lower quality.
4. Include Statistical Analysis: Provide a solid statistical analysis of the data to support your conclusions. Report data on the riboflavin content during the fermentation process, including the mean, standard deviation, and variation. To determine the importance of changes over time, think about using statistical tests.
The statistical analysis was performed and presented using the mean value and standard deviation (see Figure 2) using error bars, as we believe that the data presented in this way is more informative and representative.
5. Validate the Spectrofluorimetric Method: Carefully record the method's validation. Information on specificity, accuracy, repeatability, and limit of detection (LOD) should be included. Describe any reference standards and calibration curves that were used for validation.
We emphasize once again that this is a preliminary Communication, which we hope will correct the erroneous data (Malbaša et al. 2011) that are constantly repeated in the publications. Furthermore, now that we know the order of magnitude of B2 in kombucha, a larger study is underway that will include validation of the method, more samples and substrates.
6. Contrast with Earlier Research: Give a thorough comparison of your results with earlier research that measured riboflavin in kombucha using HPLC. Talk about the possible benefits and drawbacks of spectrofluorimetry in contrast to HPLC.
In all three papers, kombucha was grown in almost identical ways, and in both methods (HPLC, fluorimetry) excitation and emission were at almost the same wavelengths. In the discussion, we emphasized that our results are in good agreement with those published in the paper by Malbaša et al. 2004. We believe that both methods work well, but care should be taken when handling the samples and measurements should be performed immediately after sampling.
7. The riboflavin's fluorescence quantum yield: It would be advantageous to look into the riboflavin's fluorescence quantum yield in your particular experimental setup. The dependability of your results will be strengthened and your spectrofluorimetric method will be further validated by comparing your measured quantum yield with the established theoretical value.
We believe that this has nothing to do with the topic of the Communication, which focuses on solving the dilemma regarding riboflavin concentration in traditionally prepared kombucha, and that it would burden the paper.

Reviewer 2 Report
Comments and Suggestions for Authors
Find the below comments/suggestions to improve the manuscript:
- state briefly the objective, motivation, and novelty of the communication in the abstract section
- The result and discussion section are not supported with enough references to substantiate the statement
- The figures are not well discuss in the text
- Include briefly summary or concluding part to show the novelty of the work
- Updating some of the references to recent
Author Response
Thank you for your positive comments. The lack of literature data and the large differences in riboflavin concentration in almost identically grown kombucha in only two previously published papers (Malbaša et al. 2004 and Malbaša et al. 2011) inspired us to conduct this study, which is clearly stated in the abstract. These large differences in riboflavin content (3 orders of magnitude) and the results of our study prompted us to contact the author, whereupon he replied that he assumes that the high riboflavin concentrations (Malbaša et al. 2011) were the result of lysis of the cell culture due to the long storage of the samples in the freezer. Unfortunately, the results of this study are cited in numerous articles and reports published later. In the discussion, we emphasized that our results are in good agreement with those in the paper by Malbaš et al. 2004. and unfortunately there is no other data in the literature with which we could compare them. We hope that in the future the correct data (Malbaša et al. 2004 and this paper) on riboflavin content in traditionally grown kombucha, which ranges from 7.63∙10-8 to 2.2∙10-7 mol dm-3, will be cited.
Figure 1 shows the calibration curve of riboflavin, which in our opinion does not need to be discussed. The results of the statistical analysis of the B2 content, presented using the mean value and standard deviation, can be seen in Figure 2, and as this work is a Communication, we do not consider it necessary to discuss daily results.
We have added a concluding section in which we consider the novelty and scientific contribution of the work as well as some more recent references (all highlighted in yellow).
Reviewer 3 Report
Comments and Suggestions for Authors
This is a very nicely written paper. It is brief and to the point. However, I have some suggestions to improve the manuscript as follows:
1. What was the ambient temperature during the fermentation? Is the ambient temperature typical to all other studies. The reason I ask this question is, does the ambient temperature during fermentation have an effect on the B2 concentrations observed in this study, which are substantially lower than in the previous studies.
2. Not sure if the authors considered any replicates for each concentration of sucrose. Ideally, they should consider multiple replicates so that they can estimate the variance in the estimated values of B2 concentrations. This will allow them to compute 95% confidence intervals as well. Otherwise it is very hard to judge the results presented here statistically.
3. Once multiple replicates are performed, using the estimated B2 concentration along with standard errors, the authors can perform a formal statistical test comparing their results with what is published in the literature to determine if their values are statistically significantly different.
4. If the values derived by them are significantly different, then I would expect the authors to provide a scientific explanation for why their results are different. Are the experimental conditions different? Do the experimental conditions result in different microbial ecologies during the fermentation process that result in lower B2 concentrations? How different are the methods of estimation that resulted in such differences. I would appreciate a more detailed discussion along these lines.
Overall this is an interesting project but needs more details.
Author Response
Thank you for your positive comments. Fermentation was carried out at room temperature in all three studies (Malbaša et al. 2004, Malbaša et al. 2011, this study). The lack of literature data and the large differences (3 orders of magnitude) in riboflavin concentration in almost identically cultivated kombucha in two previously published papers (Malbaša et al. 2004 and Malbaša et al. 2011) inspired us to conduct this investigation. We assumed that the riboflavin content in kombucha is about 10-4 mol dm-3 as stated in the paper by Malbaš et al. 2011 and later cited in numerous other papers. When we got the results of our study, which were in good agreement with those of Malbaša et al. 2004, we contacted the author, he replied that he assumes that the high riboflavin concentrations (Malbaša et al. 2011) were the result of lysis of the cell culture due to the long storage of the samples in the freezer. Therefore, we decided to write this study in the form of a Communication in the hope that the correct data (Malbaša et al. 2004 and this paper) on riboflavin content in traditionally grown kombucha will be cited in the future.
Since this preliminary study is the only one besides Malbaša's that deals with B2 concentration in traditionally grown kombucha, we are certainly planning a much more extensive study in the future, in which we would do everything you suggest.
Round 2
Reviewer 1 Report
Comments and Suggestions for Authors
The manuscript has the potential to be published as a "Communication" but requires revisions to enhance its novelty and impact.
Revision Suggestions to Increase Impact:
Strengthen the Discussion:
Explain Variations: Discuss possible reasons for the fluctuating riboflavin content during fermentation. Consider factors like:
Microbial consumption and production of riboflavin.
Changes in pH affecting riboflavin stability.
Potential binding of riboflavin to other kombucha components.
Compare to Other Beverages: Provide context by comparing the measured riboflavin levels with other common beverages. This comparison will highlight the nutritional implications of the findings.
Author Response
Dear reviewer,
we have accepted all your suggestions and included them in the revised manuscript. Parts of the improved discussion are highlighted in red.